# Psychiatric Symptoms in Parkinson’s Disease Patients before and One Year after Subthalamic Nucleus Deep Brain Stimulation Therapy: Role of Lead Positioning and Not of Total Electrical Energy Delivered

**DOI:** 10.3390/jpm12101643

**Published:** 2022-10-03

**Authors:** Teresa Somma, Felice Esposito, Maria Rosaria Scala, Antonio Scelzo, Cinzia Baiano, Sara Patti, Vincenzo Meglio, Felice Iasevoli, Luigi M. Cavallo, Domenico Solari, Andrea De Bartolomeis, Paolo Cappabianca, Giordano D’Urso

**Affiliations:** 1Department of NeuroSciences and Reproductive and Dental Sciences, Division of NeuroSurgery, Università degli Studi di Napoli Federico II, 80131 Naples, Italy; 2Department of NeuroSciences and Reproductive and Odontostomatological Sciences, Division of NeuroSurgery, Federico II University of Naples, 80131 Naples, Italy; 3Neurosurgery Unit, Federico II Medical Center, 80131 Naples, Italy; 4Department of NeuroSciences and Reproductive and Dental Sciences, Division of Psychiatry, Università degli Studi di Napoli Federico II, 80131 Naples, Italy

**Keywords:** Parkinson’s disease, deep brain stimulation, psychiatric symptoms, subthalamic nucleus, total electric energy delivered, complications

## Abstract

Parkinson’s disease (PD) patients may experience neuropsychiatric symptoms, including depression, anxiety, sleep disturbances, psychosis, as well as behavioral and cognitive symptoms during all the different stages of the illness. Deep Brain Stimulation (DBS) therapy has proven to be successful in controlling the motor symptoms of PD and its possible correlation with the occurrence or worsening of neuropsychiatric symptoms has been reported. We aimed to assess the neuropsychiatric symptoms of 14 PD patients before and after one year of Subthalamic Nucleus (STN)-DBS and to correlate the possible changes to the lead placement and to the total electrical energy delivered. We assessed PD motor symptoms, depression, anxiety, apathy, impulsivity, and suicidality using clinician- and/or self-administered rating scales and correlated the results to the lead position using the Medtronic SuretuneTM software and to the total electrical energy delivered (TEED). At the 12-month follow-up, the patients showed a significant improvement in PD symptoms on the UPDRS (Unified Parkinson’s disease Rating Scale) (−38.5%; *p* < 0.001) and in anxiety on the Hamilton Anxiety Rating Scale (HAM-A) (−29%; *p* = 0.041), with the most significant reduction in the physiological anxiety subscore (−36.26%; *p* < 0.001). A mild worsening of impulsivity was detected on the Barratt Impulsiveness Scale (BIS-11) (+9%; *p* = 0.048), with the greatest increase in the attentional impulsiveness subscore (+13.60%; *p* = 0.050). No statistically significant differences were found for the other scales. No correlation was found between TEED and scales’ scores, while the positioning of the stimulating electrodes in the different portions of the STN was shown to considerably influence the outcome, with more anterior and/or medial lead position negatively influencing psychiatric symptoms.

## 1. Introduction

Parkinson’s disease (PD) is the second most common neurodegenerative disorder, occurring with 0.3–1% prevalence in the 60–80-year range. It is characterized, in the most common form of the pathogenesis, by dopaminergic neurodegeneration of the nigrostriatal pathway, resulting in both motor and non-motor symptoms [1,2,3,4]. Deep Brain Stimulation of the Subthalamic Nucleus (STN-DBS) proved to be highly effective in treating many of PD’s motor symptoms and often allows medication reduction [5]. However, some authors have highlighted a possible correlation between DBS and the occurrence or worsening of psychiatric symptoms [6,7,8,9,10]. In particular, mood alterations of both polarities were reported, with some patients displaying apathy, depression, and suicidality and others developing mania and impulse control disorders, with significant impairment of personal functioning in either case. To date, several prospective and retrospective studies have investigated this phenomenon, still showing conflicting results, mainly due to differences in population characteristics and study designs [6,7,8,9,10]. The present study involved a sample of PD patients and aimed to assess their psychiatric symptoms before and after one year of STN-DBS, as well as to correlate the possible psychiatric changes to the DBS leads’ placement in the different portions of the Subthalamic Nucleus (dorsolateral, medial, and anterior) and to the total electrical energy delivered (TEED) to the tissue [11].

## 2. Materials and Methods

### 2.1. IRB/Ethics Committee Approval and Informed Consent

This study reports research involving human participants. However, since it does not report any experimental treatment nor a randomized controlled trial, and the involved subjects received the standard treatment in our institutions, with no modifications of the treatment protocols, the IRB/Ethics committee approval was not needed and was not sought.

All the subjects signed a written procedure-specific informed consent, and the form was included in the patients’ charts. The written informed consent also regarded the data from patients’ medical records to be used in medical research. The study did not include minors.

After the collection, data were anonymized before the statistical analysis.

### 2.2. Patient Population

Fourteen non-consecutive PD patients underwent placement of bilateral DBS system at the Division of Neurosurgery of the Federico II University of Naples (Italy) from 2014 to 2019, with selective targeting of the dorsolateral region of the Subthalamic Nucleus (STN). The subjects enrolled in this study were not consecutive because we selected only those patients who had a complete pre- and postoperative neuropsychiatric evaluation and for whom we could obtain a reliable map of the leads’ positioning in the STN and the study of the TEED.

The selected patients were evaluated by the same team of neurologists, psychiatrists, and neurosurgeons three months before the surgery and every three months afterwards in the first year. The neurological assessment was performed on PD medication (ONmed) and after a 12 h medication withdrawal (OFFmed) using the Unified Parkinson’s Disease Rating Scale (UPDRS).

From the day following surgery, L-dopa dosage was gradually lowered to the minimum needed, according to the patients’ motor symptoms. About two weeks after the surgery, the pulse generator was activated, and, in the subsequent months, the optimal settings for all patients were reached following the outpatient clinic visits.

The demographic and clinical characteristics of the patients are outlined in Table 1. There was no difference in age at surgery and of the duration of the disease before the operation in relation with the gender.

### 2.3. Assessment

The eligibility for surgery was assessed by a multidisciplinary team of neurosurgeons, neurologists, psychiatrists, and neuropsychologists using the Core assessment program for surgical interventional therapies in Parkinson’s disease (CAPSIT-PD) [11].

At baseline, the Mini-International Neuropsychiatric Interview (M.I.N.I.) was ad-ministered to identify any pre-existing psychiatric disorder [12].

For the assessment of depression, we used two different rating scales, one clinician-administered, i.e., the Hamilton Depression Rating Scale (HAM-D) [13], and one self-administered, i.e., the Beck Depression Inventory (BDI) [14]. Score ranges for the severity of depression are: for HAM-D, 0–7 subclinical, 8–17 mild, 18–24 moderate, ≥25 severe; for BDI, 0–9 subclinical, 10–18 mild, 19–29 moderate, ≥30 severe. For anxiety, we used two different rating scales, one clinician-administered, i.e., the Hamilton Rating Scale for Anxiety (HAM-A) [15], and one self-administered, i.e., the Beck Anxiety Inventory (BAI) [16]. Score ranges for the severity of anxiety are: for HAM-A, <17 mild, 18–24 moderate, 25–30 severe, >30 persistent; for BAI, <21 mild, 22–35 moderate, >36 severe. HAM-A has a total score and, according to a two-factor model, two subscores corresponding to the different dimensions of “cognitive” (HAM-A/C) and “physiological” (HAM-A/P) anxiety.

For apathy, we used the Apathy Evaluation Scale (AES) [17] with a score range of 18 to 72, and for impulsiveness the Barratt Impulsiveness Scale-11 (BIS-11) [18], with a score range of 30 to 120. BIS-11 has a total score and, according to a three-factor model, three subscores corresponding to the different dimensions of “attentional” (BIS-11/A), “motor” (BIS-11/M), and “non-planning” (BIS-11/NP) impulsiveness.

For suicidality, we used the Scale for Suicide Ideation (SSI) [19] with a score range of 0 to 38 and the Reason for Living Inventory (RFL-48) [20] with a score range of 0 to 48 (higher scores represent more reasons to live).

All these psychiatric rating scales were administered during the 3 months preceding the surgical procedure (*T*_0_) with the patients in “ON-MED” state, approximately one hour after taking the drug. Postoperative assessments were performed 12 months after the operation (*T*_1_) in “ON-stimulation” (ONstim) condition. The administration of psychiatric scales lasted for an hour and a half for each patient at each timepoint; the presence of family members or relatives that could have influenced the patient’s responses was avoided. Subsequently, the patients’ responses were confirmed by interviews with family members and caregivers.

### 2.4. Treatment

All patients underwent the bilateral implantation of a DBS system with selective targeting on the dorsolateral region of the STN [21].

All patients were selected by the attending neurologists and evaluated by a multispecialistic team, including the neurosurgeon, the neurologist, the psychiatrist, and the neuro-anesthesiologist. Based on the PD symptoms, the team suggested the most appropriate target for the DBS therapy; in all the subjects enrolled in this study the chosen target was the STN.

The operation was commonly performed in two steps: (i) the lead positioning in the selected target using the stereotactic frame, in awake surgery using the dexmedetomidine for sedation during some parts of the procedure, and with the aid of the intraoperative neuromonitoring (either the microregistration and stimulation) for the target confirmation and for stimulation-related complication avoidance; (ii) the lead extensions and the pulse generator positioning, under general anesthesia.

Our framework for the path planning integrates multi-modal MRI (Magnetic Resonance Imaging) analysis (*T*1w, *T*2w, FLAIR) to compute suitable patient-specific DBS lead trajectories that optimize the avoidance of specific critical brain structures. The MRI images are fused with the CT (Computed Tomography) obtained after the stereotactic frame application; the tridimensional coordinates of the chosen target and the coordinates of the chosen entry point (usually in the pre-coronal, bilateral, paramedian frontal region) are obtained, which, ultimately, define the lead trajectories.

Such standard algorithm [22] has been employed in all the included patients.

### 2.5. Imaging

Preoperative planning was performed for each patient using PD-specific sequences of MRI (FLAIR, MP-RAGE, MP2-RAGE). On the first postoperative day, a CT scan of the brain, integrated with radiography extended to the right subclavian region, was performed to check the leads’ positioning and the integrity of the components of the DBS system. The images of the postoperative CT scan were fused with the images of the preoperative MRI using the SureTune™ software (Medtronic, Minneapolis, MN, USA) to obtain a detailed and tridimensional localization of the electrodes [23,24].

### 2.6. TEED Estimation

Total electrical energy delivered (TEED) per second through the STN-DBS leads was estimated using the Koss formula [25]. The stimulation parameters and the contact impedance were read at 12 months after DBS, and TEED values were calculated without side distinction.

### 2.7. Statistical Analysis

All data analyses were performed using the IBM^®^ SPSS^®^ package (version 27.0.1.0). Results are expressed as means ± SD for both discrete and continuous variables. For the assessment of within-subject changes between *T*_0_ and *T*_1_, 2-tailed paired t-tests were computed for the means of the total scores of each psychiatric scale and for the means of the two subscores of the HAM-A and of the three subscores of the BIS-11. A value of *p* ≤ 0.05 was considered significant. TEED analysis was performed using the Koss formula. The TEED values were calculated among the patients who showed a new onset or a worsening of the psychiatric symptoms versus those who did not show such postoperative complications, using parametric tests (Wilcoxon test for non-normal quantitative observations).

## 3. Results

The results of the psychiatric rating scales of the 14 patients enrolled in the study are summarized in Table 2 and Table 3. At baseline, none of them met the DSM5 criteria for major psychiatric disorders. Eight patients suffered from an adjustment disorder related to the functional impairment due to symptoms of PD. Six of them had adjustment disorder with mild depressed mood and were taking selective serotonergic reuptake inhibitors (SSRIs), while two had adjustment disorder with mixed anxiety and mild depressed mood and were taking SSRIs and benzodiazepines.

At the 12-month follow-up evaluation we found a significant improvement in PD global symptoms on the UPDRS (−38.5%; *p* < 0.001) (see Figure 1), and in anxiety on the Hamilton Anxiety Rating Scale (HAM-A) (−29%; *p* = 0.041) (see Figure 2), with the most significant reduction in the physiological anxiety subscore (−36.26%; *p* < 0.01) (see Figure 3). A mild worsening of impulsivity was detected on the Barratt Impulsiveness Scale (BIS-11) (+9%; *p* = 0.048) (see Figure 4), with the greatest increase in the attentional impulsiveness subscore (+13.60%; *p* = 0.050) (see Figure 5). No statistically significant differences were found for the other scales (Table 2).

The electrodes were located within the STN for all patients (see Figure 6); however, the patients with one or both electrodes located in a more medial and/or anterior position were those displaying the worst psychiatric outcome, i.e., an increase in impulsiveness at the BIS-11. We did not observe any significant correlation between TEED and the scores of the psychiatric scales, calculated between the subgroups of patients who reported and those who did not report a modification between the preoperative and postoperative values of the different neuropsychiatric scales (TEED 0.0293 ± 0.000741 vs. 0.0291 ± 0.00127; *p* = 0.81).

### Gender Difference Analysis

As there is evidence of gender difference in psychiatric symptoms in Parkinson’s disease [30,31,32,33], we also performed the analysis with the distribution of all scores by gender.

We compared the preoperative ONmed and the postoperative ONstim values of all scales using the Independent-Samples Median Test as Hypothesis Test and found that there was no gender difference for any of the evaluation scales used in this study.

## 4. Discussion

DBS therapy is a well-established and effective treatment for patients with movement disorders and, particularly, with PD. However, some undesirable psychiatric outcomes have been reported after the surgical procedure and are a matter of discussion [6,7,8,9,10].

Several hypotheses have been formulated to explain the complex relationship between psychiatric symptoms and DBS for PD [34,35,36,37]. First, some PD features have been indicated as vulnerability factors, i.e., family history, pre-existing non-motor fluctuation, disability grade, and disease duration [5]. Moreover, higher levels of preoperative depression have been correlated to the worsening of psychiatric symptoms after surgery [38]. However, a clear distinction between non-motor PD symptoms and mood disorders can be very challenging. Nevertheless, the personal and family history as well as the symptom combination can help differentiate the two conditions, with apathy, reduced appetite, and early awakening in the morning being more specific to PD, whilst sadness and feelings of unworthiness and hopelessness are more typical of depression. In addition, even in the case of clear depressive symptoms, it may be not easy to differentiate a depressive reaction to the PD-related disability from an endogenous major depression. This has relevant clinical implications since, in the first case, improving the disability by means of DBS could lead to a mood improvement, while in the latter case, a pre-existing major depressive disorder could be a risk factor for more severe psychiatric outcomes after surgery.

Furthermore, each patient can present variations in the neurodegeneration process. Canesi et al. [39] reported that mood alterations in PD, considering both depressive symptoms and mood elevation, are related to the advanced stages of the disease as well as the presence of impulsive disorders, and dopaminergic therapy alone would not always be able to restore a normal mood condition. In addition, it has been described that the occurrence of psychiatric disorders is correlated with the meso-cortico-limbic dopaminergic denervation more than the nigrostriatal denervation [40,41,42,43].

Another aspect to consider, according to Thobois et al. [42], is the Delayed Dopamine Withdrawal Syndrome (DAWS). It is due to the consistent postoperative reduction in dopamine and L-dopa medications, and it is characterized by apathy, depression, and suicide attempts that have a significant percentage of regression after dopamine agonist administration. Apathy is a common feature of PD and depression [4,44,45]. In our series, apathy did not improve after treatment despite the improvement of both motor and mood symptoms. A possible explanation is that the baseline level, as from the AES mean score, was already under the cut-off for the presence of apathy in PD patients [44].

In this study, we investigated the relationship between the DBS therapy and the psychiatric effects in a sample of 14 PD patients, through a comprehensive psychometric evaluation performed before and 1 year after the DBS operation. Moreover, we focused our attention on the possible correlation between the changes in psychiatric scores and two key technical aspects of DBS therapy, i.e., the total energy delivered and the lead positioning within the STN. At the one-year follow-up, we found a reduction in anxiety, depression, and suicidality scores, with the most prominent and statistically significant improvements in the subscore of “physiological anxiety” (i.e., somatic manifestations of anxiety), which reduced by 36.26% (*p* < 0.001). We also observed a mild worsening of impulsiveness (+8.98%; *p* = 0.048).

For depression and for anxiety symptoms, we decided to use both clinician-rated (HAM-D and HAM-A respectively) and self-administered scales (BDI and BAI respectively). The reason for this was that, unlike the other psychopathological dimensions that we assessed, depression and anxiety symptoms can hardly be differentiated from apparently similar PD features. In particular, depression symptoms can hardly be differentiated from apathy, bradykinesia, and facial amimia typical of PD, while symptoms of physiological anxiety can hardly be differentiated from somatic and vegetative symptoms in the context of PD [34,46]. We hypothesized that patients could have had a different perception of this distinction compared to clinicians and that a comparison between patient and clinician ratings could have yielded extra information, particularly regarding the relationship between the patients and the illness and/or the implanted device. While no difference between clinicians’ and patients’ evaluation was detected regarding the depression symptoms, a difference was found in the case of the symptoms of physiological anxiety. In fact, a statistically significant reduction in this dimension was reported by clinicians and not by patients. This different perception is not easily interpreted. A possible explanation is that on the one hand the patients’ evaluation could have been biased by the attribution of the anxiety improvement to the overall PD improvement and, on the other hand, clinicians could have had an over-optimistic attitude towards the beneficial effects of the DBS treatment on psychiatric symptoms. Further studies with a wait-list control group and blind raters could help solve this issue.

All our patients underwent the same degree of reduction in dopaminergic therapy, but the patients could have been affected in different ways according to disease duration or degrees of motor impairment. The analysis performed between preoperative and postoperative data turned out statistically significant for two psychological tests: HAM-A (*p* = 0.0410) and BIS-11 (*p* = 0.0483). We did not find any statistical differences between patients who had a history of anxiety or mild/moderate depression and those who did not have any pre-existing psychiatric disorder. However, in two cases, we assisted a significant mood improvement. Both patients had a longer disease duration with a history of mild depression. After DBS, they experienced improvement of their disabling motor symptoms, with a significant better change in their quality of life and in their depressed mood. Furthermore, in the two patients, the mapping of the leads inside the STN showed an optimal positioning inside the dorsolateral portion of the STN, thus confirming the important role of the stimulation of such neurons in improving the motor symptoms and, ultimately, the mood disorders. This is in line with the findings of Combs et al. [37]; they reported a better chance of larger reductions in depressive symptoms correlated with the length of time an individual lives with Parkinson’s disease prior to seeking DBS, hence with longer disease duration. This aspect is even more important to consider since patients with high levels of depressive symptoms are routinely assessed as weaker candidates for surgical consideration [43].

Some studies implied that an initial improvement in anxiety might be ascribable to the improvement in the motor domain, but it can be followed by a worsening subsequent to the several adjustments of stimulation parameters that the patients have to endure in the postoperative period (particularly, an increase in voltage and intensity) or alteration of limbic circuits [41], whereas other studies did not find any significant correlation between DBS and anxiety [6].

The worsening of impulsivity has been poorly reported [42,44]. However, albeit in our series the BIS-11 scores were statistically increased at follow-up, they still remained within the normal range of both the general and the PD population [47]. Furthermore, the intake of dopamine agonists in PD patients is a risk factor for the development of an impulsive disorder [48].

In this scenario, it is important to underline that the increase in impulsive behaviors is in close relationship with the physiological role of the STN [9]. Indeed, it is involved in corticocortical motor networks, but also in cognitive and emotional pathways [9]. In particular, the ventromedial region actively draggers in the limbic circuits. It appears essential to allow the integration of all information related to the decision-making process for preventing premature, impulsive responses, especially in high-conflict situations. In other words, it is referred to a «hold your horses» signal, allowing to delay the decision and gather additional information to choose the best option [49].

Thus, leads’ positioning within the different subregions of the STN can play a role in the mood disorders, induced by direct and indirect stimulation of the pathway enrolled in the mood control. York et al. [50] support the hypothesis that the dorsolateral subthalamus, responsible for motor control, is the most suitable site for electrode placement, since it guarantees the best outcome. It is well established that the STN may be functionally divided into three areas: motor, limbic, and associative. The motor pathways are located in the dorsolateral region, where neurons are further organized in a somatotopic pattern. In this contest, the posterior area is populated by motor neurons directed to muscles of the trunk, ultimately responsible for postural improvement and motor changes. Therefore, the stimulation of this portion of STN can cause positive effects on motor impairments and after DBS, positive impacts on the mood and psychoaffective changes. This fact has been already reported in previous studies. Furthermore, in order to objectively study the lead position in the STN nuclei without the possible confounding of effects of the artifacts generated by the leads in the images of the MRI, we used the Medtronic SureTune™ software, ensuring a detailed and tridimensional evaluation of the leads’ localization, and the map of the lead positioning is shown in Figure 6. Indeed, our patients experienced an improvement of motor symptoms with no significant changes in mood status. As already said, two cases with more important and early improvement of their preoperative although slight psychiatric symptoms had a more lateral position of the contacts, confirming the key role of the lead location for the clinical outcomes. As a matter of fact, patient #13 experienced a postoperative favorable neuropsychiatric performance, and their lead positions were found to be in the dorsolateral portions of the STN. On the other hand, patient #10, who experienced a slight worsening of the postoperative neuropsychiatric performance, had the left lead placed in the center of the STN and the right lead in the medial portion (see Figure 7).

Another aim of our study was to try to correlate the psychiatric symptoms and the quantity of energy delivered (TEED). To the best of our knowledge, this correlation had never been specifically investigated before, whereas few studies had focused on the relationship between TEED and motor outcomes [41], and in only one very recent report, energy delivered was correlated with personality trait shifts. In terms of stimulation settings, constant current (CC) and constant voltage (CV) to achieve equivalent motor efficacy have not shown any significant differences in non-motor outcomes, including cognition, mood, and quality of life. Dayal et al. in their review [51] analyzed the side effects emerging as a result of a change in the stimulation parameters delivered through a DBS contact. However, because of the natural anatomical variation between patients, as well as variations in surgical technique, targeting, and precision, there is inevitably a confounding factor in the interpretation of stimulation adjustments between patients. Our TEED analysis showed no correlation with the observed changes in psychiatric symptoms.

A secondary end-point of this study was to define if there was any gender difference in psychiatric symptoms in our patient series. As a matter of fact, it is established that either motor and non-motor symptoms in patients with PD show differences related with gender [30,31,32,33]. For such aim, we compared the preoperative ONmed and the postoperative ONstim values of all scales used in this study and found that there was no gender difference for any of the evaluations. As a result, we had to retain the null hypothesis that, at least in our series of seven males and seven females, either the preoperative or the postoperative motor and psychiatric symptoms were of the same grade in the two groups. This may be due to either the relatively small patient series or to the fact that the patients undergoing DBS therapy surgery are a well-selected and homogeneous subgroup of subjects if compared to a more general PD population.

Given the complexity of the pathology and the limitations that underline a correct neuropsychiatric evaluation, further studies with a more extensive series are needed, perhaps collecting data from multicenter studies. We believe that, with the widespread use of directional leads and of closed-loop systems that adjust the stimulation parameters according to biomarkers, which reflect the patients’ clinical state (e.g., BrainSense technology or AlphaDBS technology), it will be possible to further personalize the stimulation therapy and lower the incidence of stimulation-related adverse effects [52,53].

## 5. Study Strengths and Limitations

To our knowledge, few studies have assessed psychiatric symptoms of PD patients before and after STN-DBS therapy and even fewer aimed to correlate the possible psychiatric changes to the DBS leads’ placement in the different portions of the Subthalamic Nucleus (dorsolateral, medial, and anterior) and to the total electrical energy delivered (TEED) to the tissue [11]. In doing so, our work could provide useful information for the current debate on the putative psychiatric and psychological changes induced by DBS [54].

Despite being one of the few of this kind, the present study has some limitations that hinder the generalization of the results. First, the relatively small sample size may have led to type I error both in the statistical analysis and in our anatomo-clinic inferences. Second, the relatively high rate of patients taking antidepressants and the fact that they were nonconsecutively enrolled make our sample likely not representative of the population of PD patients undergoing DBS. Third, we only have two time points, i.e., baseline and one year after the surgery. The lack of intermediate assessments impeded the identification of the time course of the observed effects and made it more difficult to detect potential life events that could have biased the outcomes. In order to overcome this issue, it is recommendable that future similar studies would include a patient-reported measure that explicitly refers to the effect of DBS (e.g., “are you happy with the DBS one year after the operation?”).

Moreover, considering the potentially life-long duration of DBS therapy, we cannot rule out that the observed outcomes are transient and not representative of the real impact of DBS on the psychopathological condition of patients.

Finally, the baseline psychiatric scores of the patients in this sample were in the range of either mild severity (HAM-D, BDI, HAM-A, BAI) or normality (AES, BIS-11, SSI, RFL-48). Therefore, it is questionable whether the observed improvements, even if statistically significant, have a real clinical relevance and could also be obtained in more severe clinical pictures.

## 6. Conclusions

Our findings indicate that a tailored and multidisciplinary patient evaluation, alongside a correct target selection, as well as the maximal accuracy in the postoperative treatment, ensure the best efficacy and safety of Deep Brain Stimulation with selective targeting in the STN for Parkinson’s disease. Our data, even though coming from a relatively small series, indicated that, as for motor symptoms of PD, the non-motor symptoms regarding the neuropsychiatric performance are also influenced by the positioning of the leads in the different portions of the STN, and a great effort should be used in avoiding the lead positioning in the limbic and/or the associative portions of the STN. In particular, the onset and/or the worsening of psychiatric symptoms can be avoided if a thorough baseline psychopathological assessment excludes major psychiatric disorders. Furthermore, the most important single factor ensuring both a good motor outcome and the lack of psychiatric unwanted effects seems to be the lead placement in the dorsolateral portion of the STN.

## Figures and Tables

**Figure 1 jpm-12-01643-f001:**
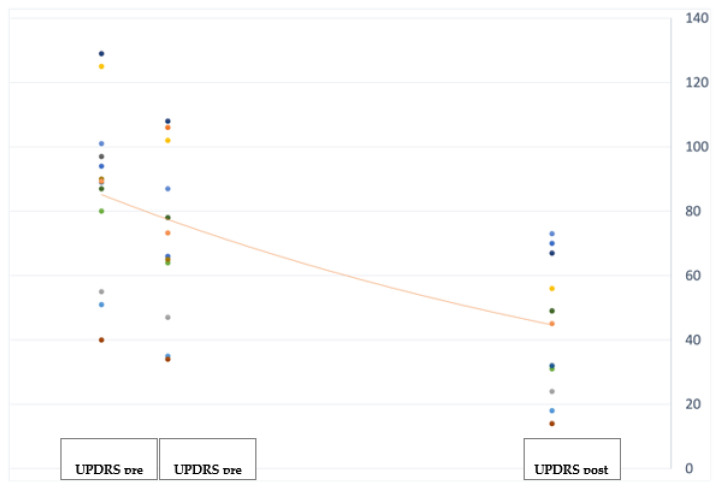
Jittered scattered plot showing the preoperative OFFmed and ONmed (**left**) and postoperative ONstim (**right**) UPDRS values of the 14 patients in the study. The difference between the preoperative OFFmed and ONmed values was not significant (*p* = 0.11), while the difference between the preoperative ONmed and postoperative ONstim showed a *p*-value = 0.00221. The line shows the negative trend of the values.

**Figure 2 jpm-12-01643-f002:**
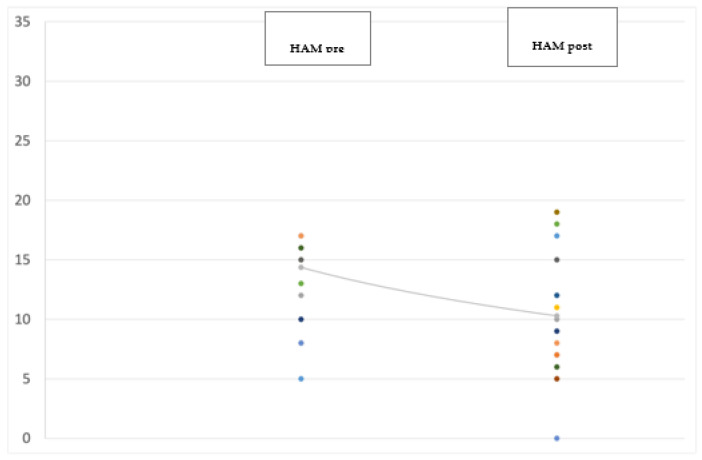
Jittered scattered plot showing the preoperative (**left**) and postoperative (**right**) Hamilton Anxiety Rating Scale values of the 14 patients in the study. The difference between the preoperative ONmed and postoperative ONstim showed a *p*-value = 0.041. The line shows the negative (improving) trend of the values.

**Figure 3 jpm-12-01643-f003:**
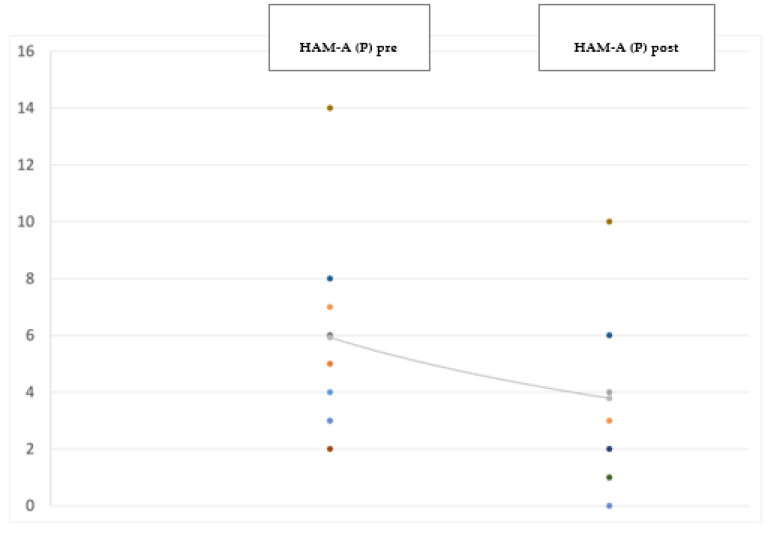
Jittered scattered plot showing the preoperative (**left**) and postoperative (**right**) physiological anxiety subscore of the Hamilton Anxiety Rating Scale. The difference between the preoperative ONmed and postoperative ONstim showed a *p*-value = 0.001. The line shows the negative (improving) trend of the values.

**Figure 4 jpm-12-01643-f004:**
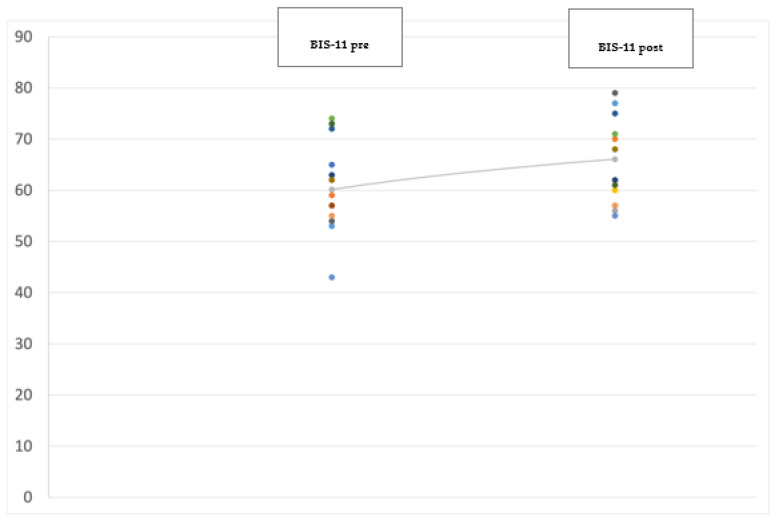
Jittered scattered plot showing the preoperative (**left**) and postoperative (**right**) Barratt Impulsiveness Scale-11 of the 14 patients in the study. The difference between the preoperative ONmed and postoperative ONstim showed a *p*-value = 0.048. The line shows the positive (worsening) trend of the values.

**Figure 5 jpm-12-01643-f005:**
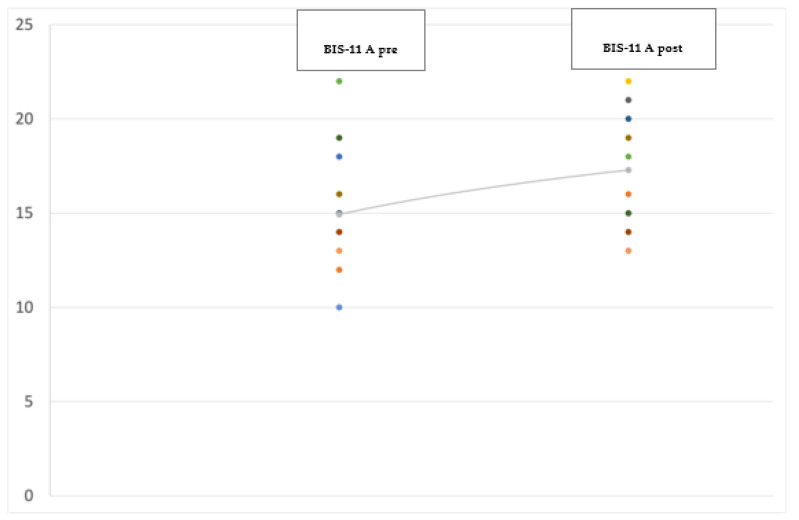
Jittered scattered plot showing the preoperative (**left**) and postoperative (**right**) attentional impulsiveness subscore of the BIS-11. The difference between the preoperative ONmed and postoperative ONstim showed a *p*-value = 0.050. The line shows the positive (worsening) trend of the values.

**Figure 6 jpm-12-01643-f006:**
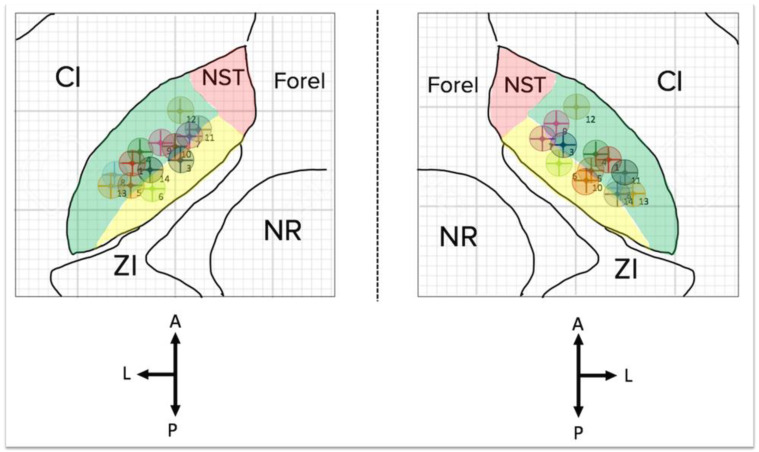
Map of the lead positionings of the 14 patients inside the SubThalamic Nucleus, bilaterally. NST: SubThalamic Nucleus (Nucleus SubThalamus); CI: internal capsule; ZI: zona incerta; NR: *Nucleus ruber* (red nucleus) of Luys; A: anterior; P: posterior; L: lateral. Drawn following [26,27,28,29].

**Figure 7 jpm-12-01643-f007:**
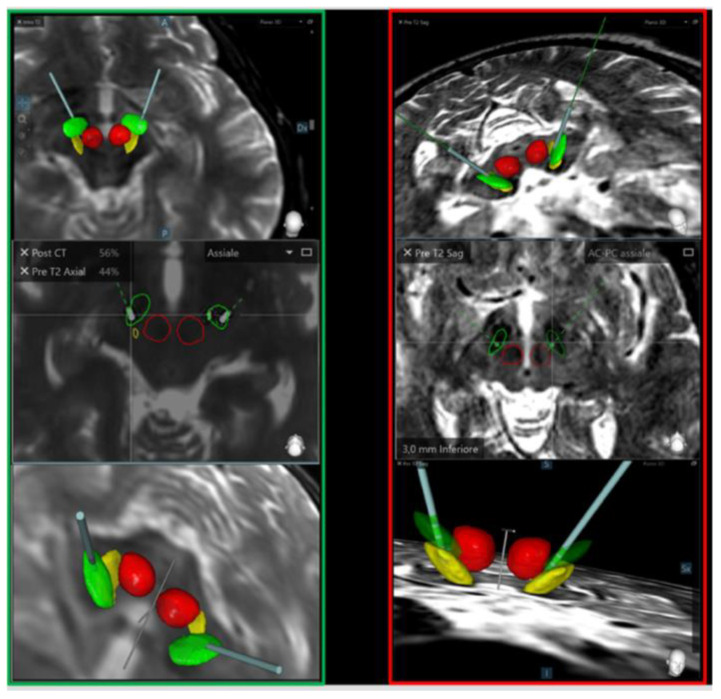
A 3D reconstruction of the lead placement inside the STN in two cases of our series. On the left is an optimal placement in the dorsolateral portion of the STN in a patient (case #13) who showed an improvement of either the motor or the psychiatric performance after the DBS. On the right it is shown the placement of the electrodes in a more medial position (especially the left electrode) in a patient (case #10) who experienced a postoperative mild worsening of the neuropsychiatric performance. Green areas: Subthalamic Nucleus; red areas: red nucleus (nucleus ruber); yellow areas: substantia nigra. The bars in light blue or in green dotted lines indicate the electrodes.

**Table 1 jpm-12-01643-t001:** Main demographics of the patients.

Subjects (*n*)	14 (7 males, 7 females)
Sex (M:F)	1:1
Age (Years)	55.7 ± 9, 12
Duration of Disease (Years)	9.78 ± 2, 51

**Table 2 jpm-12-01643-t002:** Results of the psychiatric rating scales.

Test	Mean ± SD		% Change	*p*-Value
	*T* _0_		*T* _1_	
HAM-D	10.64 ± 6.11	8.78 ± 6.67	−17.49	0.424
BDI	12.64 ± 5.27	10.5 ± 6.58	−16.93	0.285
HAM-A	14.36 ± 6.40	10.29 ± 5.46	−28.35	0.041
HAM-A (C)	8.43 ± 4.18	6.50 ± 3.13	−22.90	0.211
HAM-A (P)	5.93 ± 3.07	3.78 ± 2.72	−36.26	<0.001 *
BAI	15 ± 6.67	11.71 ± 8.79	−21.33	0.171
AES	29.35 ± 6.68	29.71 ± 6.68	1.22	0.878
BIS-11	60.14 ± 8.72	66.07 ± 8.76	8.98	0.048 *
BIS-11 (A)	14.93 ± 3.29	17.28 ± 3.34	13.6	0.05 *
BIS-11 (M)	20.64 ± 2.98	22.93 ± 3.75	9.99	0.081
BIS-11 (NP)	24.57 ± 4.99	25.86 ± 4.15	4.99	0.433
SSI	0.5 ± 1.87	0.36 ± 0.63	−28	0.800
RFL-48	4.64 ± 0.47	4.47 ± 0.35	−3.66	0.251

*T*_0_: before the DBS operation; *T*_1_: 1 year after the DBS operation; * statistically significant.

**Table 3 jpm-12-01643-t003:** Outlook of the results of the psychiatric scales, lead positioning, and TEED.

Case	Total UPDRS	HAM-D	BDI	HAM-A	HAM-A (C)	HAM-A (P)	BAI	AES	BIS-11	BIS-11 (A)	BIS-11 (M)	BIS-11 (NP)	SSI	RFL-48	Lead	Positioning	TEED
	∆ %	∆ %	∆ %	∆ %	∆ %	∆ %	∆ %	∆ %	∆ %	∆ %	∆ %	∆ %	∆ %	∆ %	left	right	
1	−37.2	−60	33.3	−58.8	−63.6	−50.0	−17.6	33.3	18.5	16.7	4.3	33.3	/	0.9	PL	PL	0.0301
2	6.1	−50	73.3	−56.25	−54.5	−60.0	−30.0	18.2	18.6	33.3	13.6	16.0	/	−14.5	PL	PL	0.0288
3	−36.8	109.1	−43.75	−16.7	20.0	−42.9	100.0	85.0	−1.8	−6.7	−5.0	4.5	/	−17.1	M	C	0.0298
4	−48.9	−46.1	−50.0	−26.7	0	−57.1	−64.7	10.0	9.1	37.5	11.8	−13.6	/	−5.7	PL	PL	0.0304
5	−37.2	−47.8	27.8	−40.625	−50.0	−28.6	−30.3	20.0	9.7	18.8	15.8	0	/	−0.9	PL	C	0.0277
6	−48.6	216.7	100.0	28.5	140.0	−25.0	100.0	2.4	−4.1	−18.2	33.3	−19.4	/	−13.9	M	M	0.0286
7	−51.6	−57.1	−36.4	−10.0	0	−33.3	−90.0	−33.3	−1.6	7.1	0	−7.4	/	3.1	C	MA	0.0284
8	−38.0	/	−50.0	−37.5	−33.3	−50.0	−57.1	−12.5	0	0	−9.5	9.1	/	−8.9	PL	PL	0.0304
9	−58.8	18.2	100.0	0	0	0	−16.7	−24.3	46.3	110.0	73.3	10.3	/	−12.2	L	CA	0.0291
10	−45.1	500	12.5	240.0	1000	50.0	45.5	27.0	45.3	40.0	23.8	76.5	/	−16.6	C	M	0.0295
11	−58.8	−46.7	−26.7	−29.4	−33.3	−25.0	−54.2	−16.7	4.2	33.3	16.0	−18.8	/	−1.2	CA	PL	0.0285
12	−59.0	−41.2	−12.5	−62.5	−61.5	−66.7	−56.3	19.2	−16.4	−21.1	41.2	−23.3	−100.0	13.3	LA	LA	0.0291
13	−37.2	−100.0	−89.5	−100.0	−100.0	−100.0	−53.8	−40.5	27.9	30.0	43.8	11.8	/	22.9	PL	PL	0.0279
14	−16.1	−16.7	−25.0	−52.9	−50.0	−57.1	−53.8	22.7	3.6	0	−30.4	47.4	/	10.2	P	PL	0.0306

Note—∆ (delta): percentual difference between preoperative and postoperative evaluation; PL: postero-lateral; M: medial; C: central; CA: central anterior; LA: lateral anterior; P: posterior; MA: medial anterior; TEED: total electrical energy delivered.

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
