# Peer review of "Psychiatric Symptoms in Parkinson’s Disease Patients before and One Year after Subthalamic Nucleus Deep Brain Stimulation Therapy: Role of Lead Positioning and Not of Total Electrical Energy Delivered"

_jpm, 2022, doi:10.3390/jpm12101643_

Round 1

Reviewer 1 Report

The paper focuses on psychiatric symptoms in patients with PD treated by STN-DBS and this is a topic of interest. The positive effect on motor symptoms of DBS is well established while the effect on psychiatric symptoms is matter of controversy, with some author suggesting even worsening of symptoms with stimulation. The authors uses a complete and detailed neuropsychiatric test-evaluation. The paper is well written and limitation of the study, in particular the limited number of sample (n=14), are well exposed. The most important suggestions of the paper are the following: a worsening of impulsiveness may follow STN-DBS; this worsening may be related to lead positioning among STN (in particular a more anterior / medial position). In particular, the latter is interesting and not previously suggested by others authors. This could be supposed due to the implication of STN (ventromedial) in limbic circuits and also cognitive-attention as well as motor control. The effect observed may be a DBS-lead position related effect and we agree with the authors about the importance of lead positioning even inside the STN.

The considerations about the possible correlation between TEED and psychiatric symptoms are of interest, especially because never investigated before.

We agree that given the complexity of the pathology, patients selection, and multidisciplinary involvment, further evaluation in future multicentric studies should be conducted.

In the final part of the discussion (Page 14) I think the following sentence lacks a conclusion:

_”We believe that with the widespread use of directional leads and of closed-loop systems that adjust the stimulation parameters according to biomarkers, which reflect the patients’ clinical state (e.g., BrainSense technology or AlphaDBS technology)”

Author Response

Reviewer #1

Please see our responses below. 

Comment 1

In the final part of the discussion (Page 14) I think the following sentence lacks a conclusion: ”We believe that with the widespread use of directional leads and of closed-loop systems that adjust the stimulation parameters according to biomarkers, which reflect the patients’ clinical state (e.g., BrainSense technology or AlphaDBS technology)”.

Response 1

We apologize for the missing conclusion of the sentence; we provided the appropriate modification.

Reviewer 2 Report

The manuscript highlighted a critical aspect of DBS therapy, normally neglected in the follow-up of patients, being a relevant study for the clinical and scientific community. However, some parts of the manuscript should improve in a relevant way, as well as more careful with the formatting of the text, it needs to be revised, parts of the text are not justified, misspellings, citation indexing, linked words, and different fonts in the text, and other aspects, showing problems in the editing of the paper

the abstract - abbreviations without the definition of the first citation, such as the UPDRS.

Introduction - it is very brief and some parts of the text are without their corresponding reference. The unique paragraph did not introduce all aspects relevant involved in this theme.  

Method - the authors did not mention the path planning for deep brain stimulation surgery, and if the same planning was used in the included patients.

- Item 2.7 is very important, and I consider that should be it the first topic of the method section. 

Results:

Figure 1 of the supplemental material did not show the representation or mention in the caption, the significant difference between data. I suggest improving the Figure with the complete statistic information and changing the box plot with the jittered data points, standardizing all figures in the same way, and also adding the variable and unit in the axis.

- I suggest including the figures of supplemental material in the body of the manuscript after it improves with the complete information in their representation. 

- the authors when describing the results do not specify which of the figures in the supplementary material they are referring to, showing a little carelessness in the transmission of information.

- In Figure 1 caption missed the definition of the NST acronym.

- Table 1 - I suggest also adding the demographic information by gender, and if there is a difference between them and change the comma to a dot in the decimal representation.

Table 2 - showed problems with the alignment of the variables

As there is evidence of gender difference in psychiatric symptoms in Parkinson's disease, I would like the authors to complement the analysis with the distribution of all scores by gender, since it is an exploratory study and the results are important for future studies and clinical direction.

The discussion section was better worked and I believe that with the inclusion of the results of the scales by gender, this discussion could be even better. Some sentences are unreferenced

Author Response

Reviewer #2

Please see our responses below. 

Comment 1

The abstract - abbreviations without the definition of the first citation, such as the UPDRS.

Response:

We thank the reviewer for having found such formal missing and we provided the definitions.

Comment 2

Some parts of the text are without their corresponding reference. The unique paragraph did not introduce all aspects relevant involved in this theme. 

Response 2

We agree with the reviewer and thank him/her for bringing this to our attention. We have integrated the references throughout the text.

Comment 3

The authors did not mention the path planning for deep brain stimulation surgery, and if the same planning was used in the included patients

Response 3

We added such information in the section 2.4 (which has been renumbered according with the comment #4).

Comment 4

Item 2.7 is very important, and I consider that should be it the first topic of the method section. 

Response 4

We thank the reviewer for this remark, and we moved the section 2.7 at the beginning of the Materials and Methods section. All the other sections have been renumbered accordingly.

Comments 5 & 6

Figure 1 of the supplemental material did not show the representation or mention in the caption, the significant difference between data. I suggest improving the Figure with the complete statistic information and changing the box plot with the jittered data points, standardizing all figures in the same way, and also adding the variable and unit in the axis. I suggest improving the Figure with the complete statistic information and changing the box plot with the jittered data points, standardizing all figures in the same way, and also adding the variable and unit in the axis.

Response 5 & 6

We followed the advice of the Reviewer we changed the chart type, also adding some statistical information.

Comment 7

I suggest including the figures of supplemental material in the body of the manuscript after it improves with the complete information in their representation

Response 7

We moved the figures from the supplemental data to the main text and renumbered them accordingly.

Comment 8

The authors when describing the results do not specify which of the figures in the supplementary material they are referring to, showing a little carelessness in the transmission of information.

Response 8

We corrected such mistake and added the reference to all figures in the text.

Comment 9

In Figure 1 caption missed the definition of the NST acronym.

Response 9

The definition of the acronym “NST” was actually present in the caption of the figure 1. It indicates the SubThalamic Nucleus (often also indicated as STN), but in its Latin form (Nucleus SubThalamus). We added such further detail.

Comment 10

Table 1 - I suggest also adding the demographic information by gender, and if there is a difference between them and change the comma to a dot in the decimal representation.

Response 10

We added such the gender count in the table. We also added in the main text that there was no difference in age at the surgery and of the duration of the disease before the operation, in relation with the gender.

Comment 11

Table 2 - showed problems with the alignment of the variables

Response 11

We corrected such problems. We feel that was due to some difference among the various word-processors used to read the text and the tables.

Comments 12 & 13

As there is evidence of gender difference in psychiatric symptoms in Parkinson's disease, I would like the authors to complement the analysis with the distribution of all scores by gender, since it is an exploratory study, and the results are important for future studies and clinical direction.

Responses 12 & 13

We that the Reviewer very much for his/her suggestion. We made such analysis and add e paragraph in the Results section as well as in the Discussion section.

Round 2

Reviewer 2 Report

The authors made all the suggested modifications, but the modified figures still have a very small axis font, which makes comprehension and reading difficult.

Author Response

We thank very much the Reviewer for having made us aware of such problem. We provided the new figures 1-5 with bigger axis fonts.